# Curvature Detection with an Optoelectronic Measurement System Using a Self-Made Calibration Profile

**DOI:** 10.3390/s22010051

**Published:** 2021-12-22

**Authors:** Christoph Thorwartl, Thomas Stöggl, Wolfgang Teufl, Helmut Holzer, Josef Kröll

**Affiliations:** 1Department of Sport and Exercise Science, University of Salzburg, Schlossallee 49, 5400 Rif, Austria; thomas.stoeggl@sbg.ac.at (T.S.); wolfgang.teufl@sbg.ac.at (W.T.); josef.kroell@sbg.ac.at (J.K.); 2Red Bull Athlete Performance Center, Brunnbachweg 71, 5303 Thalgau, Austria; 3Atomic Austria GmbH, Atomic Strasse 1, 5541 Altenmarkt, Austria; Helmut.Holzer@atomic.com

**Keywords:** accuracy, calibration profile, curvature, motion capture

## Abstract

So far, no studies of material deformations (e.g., bending of sports equipment) have been performed to measure the curvature (w″) using an optoelectronic measurement system OMS. To test the accuracy of the w″ measurement with an OMS (Qualisys), a calibration profile which allowed to: (i) differentiates between three w″ ($0.1\dot{3}$ m^−1^, 0.2 m^−1^, and 0.4 m^−1^) and (ii) to explore the influence of the chosen infrared marker distances (50 mm, 110 mm, and 170 mm) was used. The profile was moved three-dimensional at three different mean velocities ($v_{\mathrm{zero}}$ = 0 ms^−1^, $v_{\mathrm{slow}}$ = 0.2 ms^−1^, $v_{\mathrm{fast}}\;$ = 0.4 ms^−1^) by an industrial robot. For the accuracy assessment, the average difference between the known w″ of the calibration profile and the detected w″ from the OMS system, the associated standard deviation (SD) and the measuring point with the largest difference compared to the defined w″ (=maximum error) were calculated. It was demonstrated that no valid w″ can be measured at marker distances of 50 mm and only to a limited extent at 110 mm. For the 170 mm marker distance, the average difference (±SD) between defined and detected w″ was less than 1.1 ± 0.1 mm^−1^ in the static and not greater than −3.8 ± 13.1 mm^−1^ in the dynamic situations. The maximum error in the static situation was small (4.0 mm^−1^), while in the dynamic situations there were single interfering peaks causing the maximum error to be larger (−30.2 mm^−1^ at a known w″ of 0.4 m^−1^). However, the Qualisys system measures sufficiently accurately to detect curvatures up to $0.1\dot{3}$ m^−1^ at a marker distance of 170 mm, but signal fluctuations due to marker overlapping can occur depending on the direction of movement of the robot arm, which have to be taken into account.

## 1. Introduction

Different measurement principles can be applied to detect the movement of humans and objects in the three-dimensional space. Electromagnetic [1,2,3,4,5,6,7,8] and ultrasonic localization systems [9] and image processing systems (IMS) [10,11,12,13,14] are the main principles. The only local position measurement technology that reaches higher accuracy compared with these three systems are optoelectronic measurement systems (OMSs), and thus they are often considered as “gold standard” in motion capture [13,15]. OMS is based on stationary cameras and therefore can only capture data in a restricted area [16]. To estimate the 3D position of the object (i.e., marker), the light detected by the OMS is calculated via time-light triangulation.

Accuracy specifications of OMS are commonly given for markers at fixed distances from each other, moving in the observation volume. For the accuracy assessment a Standard Assessment of Motion System Accuracy (SAMSA) device or a 3D fusion FaroArm digitizer (FARO technologies Inc., Lake Mary, Florida, USA) can be used to determine reference mark distances [17,18]. In addition to a rigid bar [19] and a rotary table [20], a custom-made calibration and measurement robot was also employed for accuracy and precision analysis [21]. The relative accuracy of OMS was shown to be between 0.01 mm and 0.77 mm, depending on the capture system and experimental setup [15,19,20,21,22]. However, the marker position accuracy is usually inversely proportional to the acquisition volume, which requires that accuracies must always be viewed in terms of capture volume [18].

Studies on deformations of sport equipment (e.g., ski and snowboard bending) in combination with OMSs are scarce [23]. Specifically, no one has yet analyzed the curvature (w″) of objects using a 3D motion capture system and, consequently, no accuracy data is available. The w″ seems to be an important parameter of Alpine skis [24], and can be measured by a high-precision laser measurement system in the laboratory. Both the bending stiffness curve [23,25,26,27,28,29] and the segmental ski deflection [24] can be described based on w″. For the three-dimensional detection of w″, a laser measuring system has limited suitability, because a very complex setup with several laser units and measuring slides is required. However, such a setup would be very inflexible, since w″ could only be detected during defined three-dimensional movements, so an alternative is being investigated. Not only the curvature of Alpine skis, but also other measurement object deformations such as those of snowboards, cross-country skis, jumping skis or golf clubs could be analyzed with OMS. Another application can be the calibration but also the reliability and validity testing of bending sensors, both in three-dimensional static and dynamic settings.

Therefore, the objective of the technical note is to evaluate the capabilities and accuracy of an OMS system with respect to w″ under static and dynamic conditions. For this purpose, a self-made calibration profile had to be built.

## 2. Materials and Methods

### 2.1. Development of a Calibration Pofile with Defined Curvatures

A self-designed and fabricated calibration profile was used to determine the accuracy of OMS in terms of curvature determination (Figure 1a). The profile was designed with the program Solid Edge (Siemens PLM Software, 5800 Granite Pkwy # 600, Plano, TX 75024, USA) and milled on a high precision computerized numerical control machine (Type VMX 42 i, HURCO Werkzeugmaschinen GmbH, Gewerbestraße 5a, 85,652 Pliening, Germany). M4 countersunk screws with a length of 12 mm were screwed into the 36 threaded holes of the calibration profile. The infrared markers with a diameter of 6.5 mm and an internal thread were screwed onto the calibration profile, which clearly defines the position of the markers (Figure 1b). A circle with a known w″ from the construction is determined by three markers each. In total three w″ ($0.1\dot{3}$ m^−1^, 0.2 m^−1^ and 0.4 m^−1^) are distinguished at three different marker distances (50 mm, 110 mm and 170 mm). With a marker distance of 50 mm, the two outer markers are 100 mm apart.

The construction drawing with the corresponding tolerance specifications can be found in Figure A1. Based on the tolerance specifications, the theoretically possible maximum systematic bias was estimated (Figure A2) and presented in Table A1. It should be noted that the tolerance specifications are derived from experience and not deterministically, therefore the specifications represent the worst-case bias.

### 2.2. Accuracy Determination

#### 2.2.1. Experimental Setup

The calibration profile described in 2.1 was used to determine the accuracy of the curvature detection. For this study, an OMS with an active camera system from Qualisys (Oqus 7+, Qualisys AB, Kvarnbergsgatan 2, 411 05 Göteborg, Sweden) with an acquisition frequency of 240 Hz was employed. On an industrial articulated arm robot (IRB 6400R, ABB AG, Brown-Boveri-Strasse 5, 8050 Zurich, Switzerland) a ski was mounted on the robot arm using a test shoe (Figure 2).

The calibration profile was fixed on the ski and then the ski with the calibration profile was moved at three different mean velocity levels ($v_{\mathrm{zero}}$ = 0 ms^−1^, $v_{\mathrm{slow}}$ = 0.2 ms^−1^, $v_{\mathrm{fast}}$ = 0.4 ms^−1^) cyclically in space. The movement of the calibration profile is described by the movement of the robot coordination system (x_R_, y_R_, z_R_) and is performed as follows: (1) vertical up and down movement in z_R_-direction; (2) rotation around x_R_-axis (ski tip pointing towards the ground); (3) rotation around x_R_-axis in opposite direction (ski tip pointing upwards); (4) lateral movement in x_R_-z_R_-direction (to the left) followed by translational movement in z_R_-direction (this would correspond to edging the ski on the right edge) and (5) lateral movement in x_R_-z_R_-direction (to the right) followed by translational movement in z_R_-direction (this would correspond to edging the ski on the left edge). A total of three cycles were completed, whereby the cycle time was determined by the speed of the robot arm. Eight infrared cameras were placed around the observation area to detect the marker positions (Figure 3a). As shown in Figure 3a, 24 infrared markers were labeled with different colors, each color according to a specific curvature (orange: $0.1\dot{3}$ m^−1^, green: 0.2 m^−1^, yellow 0.4 m^−1^). The volume to be captured was about 3 × 2 × 1.5 m.

#### 2.2.2. Data Processing and Statistical Analysis

A circle in three-dimensional space is clearly defined by three points and can be described by the following parameter equation:(1)$$C:\;\vec{x}=\vec{M}+R\cdot \cos\left(t\right)\cdot \vec{u}+R\cdot \sin\left(t\right)\cdot \vec{v}$$

In Equation (1), $\vec{M}$ is the location vector of the center, R corresponds to the radius, t is the free parameter of the parameter form and is between 0 and 2π, and the orthogonal unit vectors $\vec{u}$ and $\vec{v}$ lie in the plane of the circle. The w″ can be calculated by the reciprocal of the R and is therefore uniquely defined by Equation (2):(2)$$w^{\prime \prime }=\frac{1}{R}$$

To determine the performance of the Qualisys system in terms of curvature calculation firstly the difference between the known w″ of the calibration profile and the detected w″ from the Qualisys system across all data frames i were calculated (Figure 4). Secondly, for each measurement situation ($v_{\mathrm{zero}},{\;v}_{\mathrm{slow}},\;v_{\mathrm{fast}})$ the average difference, standard deviation (SD), and maximum error were determined across the entire data points. Since the SD depends on the number of measurement points, the data series recorded with $v_{\mathrm{slow}}$ was first down sampled, consequently each data series contains three cycles with the same number of data points (N = 6117). All metrics were calculated using MATLAB (R2018B, MathWorks, Natick, MA, USA).

## 3. Results

Figure 5a shows the average difference ± SD of captured w″ from the Qualisys system and defined w″ from the calibration profile in static situation ($v_{\mathrm{zero}}$). Figure 5b also indicates the average error with the SD, but at $v_{\mathrm{fast}}$. The SD at $v_{\mathrm{zero}}$ is smaller compared to the dynamic situation. The average differences are nearly equal at $v_{\mathrm{zero}}$ and $v_{\mathrm{fast}}$, but the SDs in the dynamic situation are much higher and increase with smaller marker distances. The average difference is not greater than 138.6 mm^−1^ at a marker distance of 50 mm, resulting in a relative difference of 34.65% (with respect to 0.4 m^−1^). In contrast, at a marker distance of 170 mm, the average difference is less than −3.8 mm^−1^, which is equivalent to a relative difference of −0.95% related to 0.4 m^−1^.

Table 1 summarizes all calculated metrics and differentiates with respect to defined w″ (0.4 m^−1^, 0.2 m^−1^, $0.1\dot{3}$ m^−1^), marker distance (50 mm, 110 mm, 170 mm) and velocity ($v_{\mathrm{zero}},{\;v}_{\mathrm{slow}},{\;v}_{\mathrm{fast}})$ and shows the average difference, the corresponding SD and the maximum error. It can be noticed that the maximum error mostly decreases with larger marker distances. At a marker distance of 50 mm the maximum error is 501.5 mm^−1^ and at 170 mm the value reduces to −30.2 mm^−1^.

Figure 6 shows an example of the curvature signals calculated by the Qualisys system at different marker distances (50 mm, 110 mm, 170 mm). The curvature known from the calibration profile was 0.2 m^−1^ for all signals. First, it can be seen that at larger marker distances both the systematic error and the maximum error are reduced, and second, a periodic signal pattern can be identified.

## 4. Discussion

For this study, a calibration profile was manufactured and since the relative marker positions do not change due to the movement of the profile in space, the w″ known from the construction can be compared with the w″ calculated from the Qualisys data. The average difference between the known w″ of the calibration profile and the determined w″ of the Qualisys system, the SD and the maximum error were analyzed.

It was shown that no valid w″ can be measured at marker distances of 50 mm and only conditionally at 110 mm. A marker distance of 170 mm is sufficiently large for valid curvature detection, since the average difference between the defined and the captured w″ was less than 1.1 mm^−1^ (0.83% with respect to $0.1\dot{3}$ m^−1^) in the static situation and not greater than −3.8 mm^−1^ (−0.95% with respect to 0.4 m^−1^) in the dynamic situations. From Figure 5b it can be seen that at $v_{\mathrm{zero}}$ the SD is very small over the entire data set (with one exception at w″ =$\;0.1\dot{3}$ m^−1^ and a marker distance of 50 mm) and becomes successively smaller with larger marker distance in the dynamic situation. The maximum error also decreases with larger marker distance from 501.5 mm^−1^ at 50 mm, over 87.7 mm^−1^ at 110 mm and −30.2 mm^−1^ at 170 mm. From the bias specifications of the calibration profile given in Table A1, it can be seen that at a marker distance of 170 mm, the maximum constructed w″ bias is less than 1.4 mm^−1^ (1.05%) for a w″ of $0.1\dot{3}$ m^−1^ and less than 1.4 mm^−1^ (0.35%) for a w″ of 0.4 m^−1^. The average difference between the known w″ from the calibration profile and the captured w″ in the static situation does not exceed 1.1 mm^−1^, which is comparable to the maximum possible bias of the calibration profile. In contrast, at a marker distance of 50 mm, the average difference between the known and the captured w″ at $v_{\mathrm{zero}}$ is more than seven times larger than the maximum possible systematic bias (Table A1) of the calibration profile. However, the systematic average difference at small marker distances is consequently attributable to the measurement accuracies of Qualisys and not to the variance in the construction of the calibration profile.

A transfer of these results to other experimental settings is only possible to a limited extent, since the curvature detection with an OMS is multifactorially influenced. Marker position accuracy is typically inversely proportional to the acquisition volume of the OMS [18], thus reduced w″ accuracies are expected with larger volumes. Furthermore, the distance between the cameras and the markers, the position of the cameras relative to each other, the type and number of markers and the movement of the markers within the capture volume play a decisive role [15,20]. However, the measurement with OMS is influenced by many parameters, so that the source of the error and the reason for the deviation cannot be clearly assigned, but logical deductions can be drawn. First, the signal and the corresponding fluctuations can be considered as a function of the standardized motion of the robot arm, since a periodic pattern can be identified (Figure 6). The maximum errors at $v_{\mathrm{slow}}$ and $v_{\mathrm{fast}}$ are attributable to the acceleration and deceleration processes of specific swivel movements of the robot arm in the x_R_-z_R_-direction with simultaneous rotation of the robot arm. In this motion sequence an enhanced overlapping of the markers can be seen in some cameras, this is probably the reason for the deviations. In the static situation, these recurring fluctuations are not visible, which is also reflected by the much smaller standard deviation. However, the fluctuations could possibly be reduced by repositioning the cameras. Second, markers do not constitute spatial points but objects with finite extension, it is assumed that the centers of the spatial objects are detected by the Qualisys system. Consider a hypothetical room with a dimension of 15 × 15 m and fit a circle with a radius of 7.5 m (=w″ of $0.1\dot{3}$ m^−1^) into this room and place three markers (M_1_, M_2_, M_3_) with a distance of 50 mm on this circle. The circle and the corresponding curvature would be clearly defined and could be reconstructed on the basis of these three markers. If the middle marker M_2_ were shifted 167 µm towards the center of the circle, the markers M_1_, M_2_ and M_3_ would lie on a straight line and the curvature would be zero and the radius infinite. This example visualizes which small distances Qualisys would have to differentiate in order to detect the correct curvature at small marker distances. It seems to be quite plausible that at marker distances of 50 mm and 110 mm no or only partially valid values can be detected. However, curvature detection with an OMS is more accurate the smaller the markers are, in order to better approximate the center points. Therefore, the smallest markers available from the manufacturer with a diameter of 6.5 mm were used, but the finite size of the markers is still a source of error.

In summary, w″ above $0.1\dot{3}$ m^−1^ at marker distances greater than 170 mm can be detected validly with Qualisys in a specific experimental setting, but it cannot be answered on the basis of this investigation if these results are applicable for larger volumes, larger markers, smaller number of cameras and for other OMSs. The authors recommend for future applications that the curvature function should fit in combination with multiple markers instead of three, which should reduce signal fluctuations and make the curvature detection more stable. Future application-oriented studies are necessary to test this assumption and to determine and extend the scope of curvature detection with OMSs.

## 5. Conclusions

A self-made calibration profile was used to investigate the accuracy of curvature detection with a three-dimensional motion capture system. Curvatures larger than $0.1\dot{3}$ m^−1^ can be detected validly, considering that the distance between markers is at least 170 mm. The smaller the distances between the markers, the lower the curvature accuracy and the larger the signal fluctuations. However, it should be considered that the inaccuracy of curvature detection is magnified for some dynamic and orientation-specific movements of the markers. As a practical application, various ski-specific deformations of Alpine skis (e.g., simulation of ski bending during a carved turn) in three-dimensional space can be systematically analyzed in the laboratory. In addition to Alpine skis [23,24], other measurement objects such as snowboards [29], cross-country skis and poles [30] or jumping skis [31] can also be utilized. Furthermore, an OMS can be used to calibrate bending sensors in both static and dynamic settings, eliminating the need for a complex and difficult-to-adjust laser measurement system in the tree-dimensional space.

## Figures and Tables

**Figure 1 sensors-22-00051-f001:**
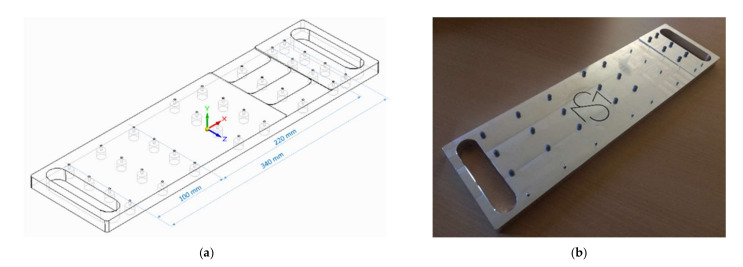
(**a**) Constructed calibration profile. (**b**) Manufactured calibration profile with M4 countersunk screws and infrared markers with a diameter of 6.5 mm.

**Figure 2 sensors-22-00051-f002:**
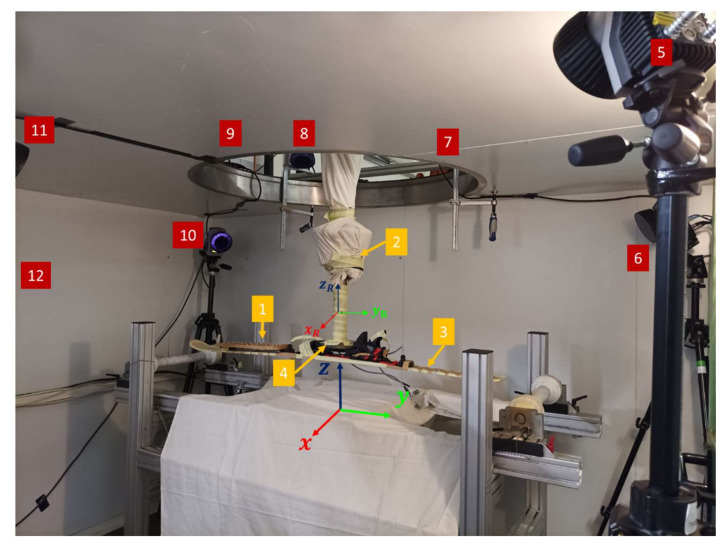
Experimental setup with the following components: 1: calibration profile with the instrumented infrared markers; 2: industrial arm robot; 3: Alpine ski on which the calibration profile was fixed.; 4: connection of robot arm with Alpine ski (test shoe); 5–12: Qualisys system with infrared cameras; the positions of the infrared markers refer to the reference coordinate system (x, y, z) and additionally a robot arm coordination system (x_R_, y_R_, z_R_) was added.

**Figure 3 sensors-22-00051-f003:**
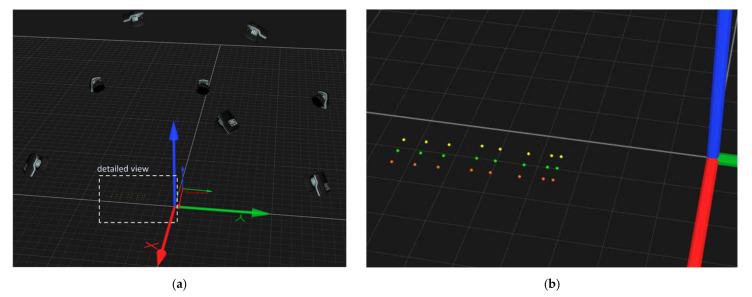
(**a**) Illustration of the experimental setup based on Qualisys Tracker Manager Software. Eight infrared cameras detecting the passive markers. The large reference coordinate system is stationary and the robotic arm coordinate system moves in space together with the instrumented infrared markers on the calibration profile. (**b**) Detailed view: The infrared markers are labeled with different colors, each color corresponding to a specific curvature (orange: $0.1\dot{3}$ m^−1^, green: 0.2 m^−1^, yellow 0.4 m^−1^). Within a color, additional distinctions are made with respect to marker distances (50 mm, 110 mm, and 170 mm).

**Figure 4 sensors-22-00051-f004:**
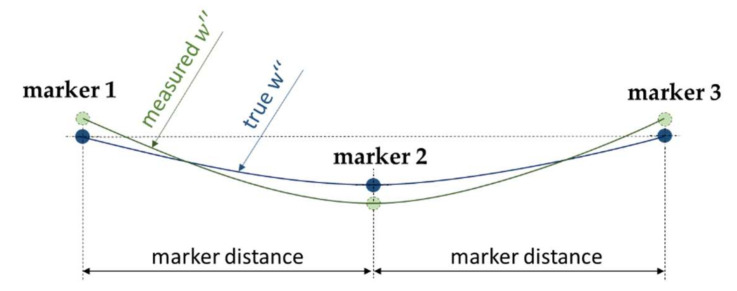
Schematic drawing at data frame i with the true marker positions in blue and the detected marker positions from Qualisys in green. Three markers define a circle with the corresponding curvature. The true w″ of the blue curve minus the measured w″ of the green curve gives the difference (in mm^−1^) at data frame i.

**Figure 5 sensors-22-00051-f005:**
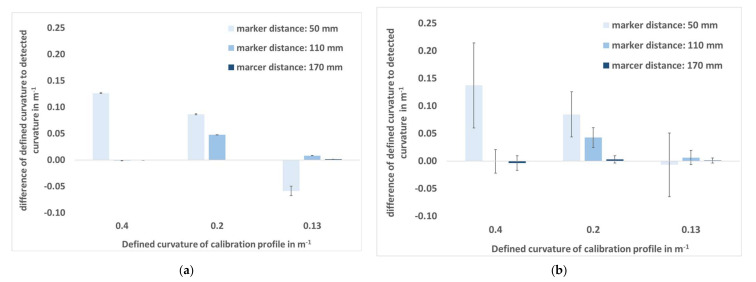
Average difference ± standard deviation (SD) in m^−1^ of captured curvature to the defined curvature. (**a**) Calibration profile at rest ($v_{\mathrm{zero}}$). (**b**) Movement of the calibration profile with mean velocity $v_{\mathrm{fast}}$.

**Figure 6 sensors-22-00051-f006:**
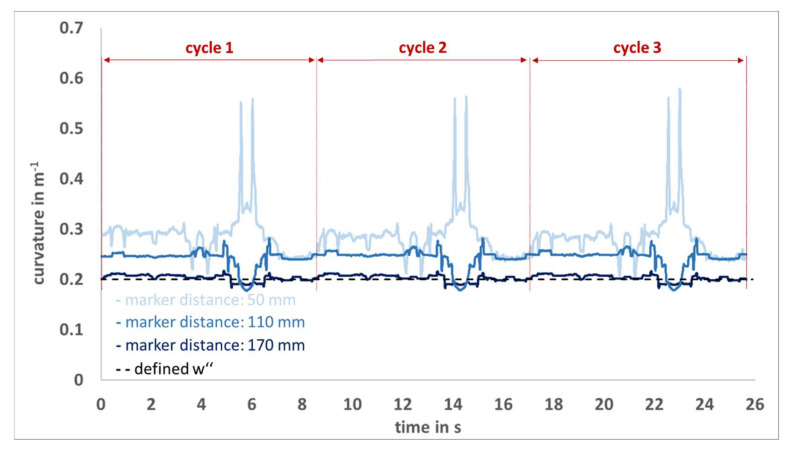
The Qualisys system signal at $v_{\mathrm{fast}}$ are shown in light blue (marker distance: 50 mm), blue (marker distance: 110 mm) and dark blue (marker distance: 170 mm). For all three signals, the defined curvature of the calibration profile was 0.2 m^−1^ (black broken line).

**Table 1 sensors-22-00051-t001:** Difference in the average calculated system measurement compared to the defined curvature from the calibration profile.

		$v_{zero}$			$v_{slow}$			$v_{fast}$		
Defined w″ (m^−1^)	Distance (mm)	Difference (mm^−1^)	SD (mm^−1^)	Max Error (mm^−1^)	Difference (mm^−1^)	SD (mm^−1^)	Max Error (mm^−1^)	Difference (mm^−1^)	SD (mm^−1^)	Max Error (mm^−1^)
0.4	50	126.3	1.0	129.8	138.6	79.3	501.5	137.2	77.1	481.41
0.2		86.3	0.9	89.7	83.8	43.4	451.3	84.5	41.1	379.5
$0.1\dot{3}$		−58.7	8.7	−74.9	−7.3	56.4	154.9	−7.0	57.7	154.3
0.4	110	−1.4	0.2	−2.2	−1.3	21.6	−83.9	−0.7	21.6	−61.8
0.2		47.6	0.2	48.2	42.4	18.2	87.7	42.5	18.1	81.5
$0.1\dot{3}$		8.1	0.2	8.7	5.8	12.7	−42.8	6.0	12.9	41.6
0.4	170	−0.4	0.1	−0.8	−3.8	13.1	−30.2	−3.7	13.3	27.7
0.2		−0.3	0.1	−0.6	2.7	6.5	−22.0	2.9	6.5	−17.9
$0.1\dot{3}$		1.1	0.1	4.0	1.05	4.6	−26.3	1.0	4.6	−20.7

w″: curvature; SD: standard deviation;
$v_{\mathrm{zero}}$,
$v_{\mathrm{slow}}$
and $v_{\mathrm{fast}}$: different robot arm velocities.

## Data Availability

The data presented in this study are available on request from the corresponding author.

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
