# Peer review of "Curvature Detection with an Optoelectronic Measurement System Using a Self-Made Calibration Profile"

_sensors, 2021, doi:10.3390/s22010051_

Round 1
Reviewer 1 Report
Congratulations to the authors for their work.
I consider it important to include in the manuscript a schematic figure with the robotic arm, the ski and the OMS (Qualisys); and additionally an image to illustrate the test of the accuracy of the w'' (measure the curvature).
Another point is to present the data that support the excerpt in the text: "In contrast, at a marker distance of 50 mm, the average absolute difference between the known and the captured w’’ at vzero is more than seven times larger than the maximum possible systematic bias of the calibration profile".
Author Response
Please see attached document for feedback.

Reviewer 2 Report
The paper describes a solution for determining the accuracy of curvature measurement (w'') using a OMS based 3D marker detection system from Qualisys. The solution is based on a self-constructed and -implemented calibration profile.
Language and grammar are good!
Minor remarks:
152: standard deviation, not derivation
218: avoid line feed in the middle of the number
Author Response

(The authors gave the same response as above.)

Reviewer 3 Report
Dear Authors, Please find my comments and recommendations below:
1) I recommend shortening the Abstract by excluding the general information that later appeared in the Introduction.
2) Line 28 (and later in the text) : -3.8 ± 13.1? If it is correct, how is it possible that the error is three times greater than the average value? It does not seem very sensible.
3) Please increase the motivation of your research in the Introduction. It’s the last paragraph (lines 69-71). In general, this part summarizes your research problems and states the goals you are aiming for. Please, extend this part.
4) The results are clearly presented and described. However, instead of presented results, there is almost no discussion. In the presented version, you mostly demonstrate the results and differences in the deviations and errors. Are movements of the robot arm the only reason for the errors? From the discussion of the results, it seems that your system will be more precise by changing the robot arm or by tuning it. In my opinion, the manuscript lacks scientific discussion and description of errors sources or reasons. This part needs to be improved by the authors.
Author Response
The response can be found in the attached document.

Round 2
Reviewer 3 Report
Authors, thank you for providing answers to my questions and taking into account all my suggestions. In my opinion, the text was improved and suitable for publication.